# Effects of antibiotic treatment on microbiota, viral transmission and viral pathogenesis of MoMuLV ts1 infected BALB/c mice

Henry Okonta[1], Xi Cheng[2], Ritu Chakravarti[2], Joan Duggan[1]*

1 Department of Medicine, College of Medicine and Life Science, University of Toledo, Toledo, Ohio, United States of America, 2 Department of Physiology & Pharmacology, College of Medicine and Life Science, University of Toledo, Toledo, Ohio, United States of America

* joan.duggan@utoledo.edu

## Abstract

The effects of normal and altered intestinal microbiota on murine retroviral transmission via the gastrointestinal tract (GIT) are diverse. The role of orally administered antibiotic treatment (ABX) on viral transmission, GIT microbial dysbiosis and subsequent pathogenesis of Moloney Murine Leukemia virus–temperature sensitive 1 (ts1) on BALB/c mice were studied. BALB/c mice were divided into four groups:

a. **ABXts1**—Treatment/Infection;

b. **ABX**—Treatment/No infection;

c. **ts1**—No treatment/Infection;

d. **Ctrl** (control)—No treatment/No infection.

ABXts1 and ABX groups showed a significant phylogenetic shift (ANOSIM p-value = 0.001) in alpha and beta diversity comparisons for microbial community composition compared to Ctrl group. Mice in the ABXts1 and ABX groups showed megacolon compared to ts1 and Ctrl groups; ABXts1 and ts1 groups showed hepatosplenomegaly, thymus enlargement, and mesenteric lymphadenopathy compared to ABX and Ctrl groups. Ctrl group had no abnormal manifestations. ABX treatment and ts1 infection uniquely affect microbial community when compared to control: ABXts1 and ABX groups significantly reduce microbiome diversity by over 80% and ts1 group by over 30%. ABXts1 and ts1 groups' viral load and clinical manifestations of infection were comparable; antibiotic treatment did not notably affect ts1 infection. Transmission and pathophysiology of ts1 infection were not significantly altered by the microbial composition of the GI tract, but ts1 viral infection did result in microbial dysbiosis independent of antibiotic treatment.

## Introduction

The gastrointestinal tract (GIT) microbiota plays an integral and diverse role in health maintenance including metabolic, trophic and protective functions. Metabolic functions range from

---

**Data Availability Statement:** All relevant data are within the paper and its Supporting Information files.

**Funding:** The authors received no specific funding for this work.

**Competing interests:** The authors have declared that no competing interests exist.

fermentation of non-digestible diet components to yield short chain fatty acids (SCFA), maintenance of immune system homoeostasis and provision of physical barrier to exogenous pathogenic microbes [1–5]. Intestinal bacteria have also been shown to promote invasion and infectivity for certain viruses, such as poliovirus, through bacterial surface-binding polysaccharides, lipopolysaccharides and peptidoglycans [6]. The intact gut microbiome has been found to be a necessary component for the establishment of infection for certain retroviruses, such as the mouse mammary tumor virus (MMTV). In this model, antibiotic-treated mice do not transmit virus to their offspring via ingestion of MMTV-laden breast milk, and dysbiosis becomes protective [7].

Moloney Murine Leukemia Virus–temperature sensitive mutant 1 (ts1) is a gamma retrovirus. When ts1 is injected intraperitoneally into BALB/c mice 72 hours after birth, they remain asymptomatic into adulthood and retain reproductive function and the females retain the capability to transmit the virus via antepartum, intrapartum, and postpartum routes [8]. Postpartum transmission involves the ingestion of infected breast milk and results in offspring that survive longer, retain reproductive functions and express comorbidities such as generalized organomegaly in spleen, thymus, mesenteric lymph nodes (MLN) and other neoplastic changes [9]. The role played by the gut microbiome in the phenotypic expression and pathophysiology of ts1 infection is uncertain.

We examined the effect of the microbiome on ts1 viral transmission, replication and pathological manifestations using the established ts1 murine model and previously validated oral antibiotic regimen (ABX) that alter murine GIT microbiome [7,8]. The ts1 model was uniquely suited for this study since transmission of this retrovirus also occurs post-partum through the GIT from oral ingestion of infected breast milk and results in reproducible clinical symptoms, including splenic neoplasm [9]. Mice were divided as follows: a) ABXts1—treated with broad-range antibiotic cocktail (ABX) to clear the bacteria populations in the GIT and infected with ts1 virus; b) ABX—treated but not infected; c) ts1—not treated but infected; and d) Ctrl–control, not treated and not infected. Comparisons were made between these groups to examine the effects of antibiotic treatment on ts1 viral transmission, pathological manifestations and microbiome diversity.

## Method

### Animals

The following protocol was approved by the University of Toledo Institutional Animal Care and Use Committee. Timed, pregnant BALB/c mice were purchased from Charles River Laboratories. Seventy-two hours after birth, female pups were divided into four groups and housed separately. Two groups (ABXts1 and ts1) received intraperitoneal injection of 20μl of 1x10⁴ FFU/ml ts1 each (Infection) and the other two groups (ABX and Ctrl) received 200μl fortified Dulbecco's Modified Eagle Medium each (No infection). Male pups were euthanized according to American Veterinary Medical Association (AVMA) panel on euthanasia. As per their guidelines, ten or less male pups under 7 days of age were placed in zippered plastic bags, fill with carbon dioxide, allow some extra time after loss of righting reflex, then transferred Zippered bag to a thermo neutral surface and placed in the freezer as the secondary method of euthanasia. All surviving female pups were weaned on day 21 and at 8 weeks of age, were paired with control uninfected males at a ratio of 3 females to 1 male. Females were checked daily for presence of cervical plugs as an indicator of pregnancy and upon confirmation, were segregated into individually labeled cages. Two of the four groups (ABXts1 and ABX) received 200μl intragastric gavage of antibiotics regimen on days 10 and 18 of pregnancy followed by *ad libitum* provision of distilled water fortified with the same antibiotic regimen (20ml

antibiotic regimen/100ml distilled water) starting on day ten of pregnancy. The other two groups (ts1 and Ctrl) were not treated with antibiotics and received regular water *ad libitum*. The offspring of these four groups continued to receive the same category of water provided to their biological mothers after weaning and they were the subjects of the study. This ensures that the antibiotic treatment groups (ABXts1 and ABX) were generated from antibiotic treated mothers with antibiotic altered bacterial diversity. The mothers were euthanized.

a. **ABXts1**—Treatment/Infection;

b. **ABX**—Treatment/No infection;

c. **ts1**—No treatment/Infection;

d. **Ctrl** (control)—No treatment/No infection.

The antibiotic regimen was prepared as follows: Kanamycin (4mg/ml), Gentamicin (3.5mg/ml), Colistin (8500 U/ml), Metronidazole (2.15mg/ml) and Vancomycin (4.5mg/ml). Fortified distilled water provided to the mothers *ad libitum* after the initial intragastric gavage and the offspring *ad libitum* for their life span for groups ABXts1 and ABX was prepared by adding 20ml antibiotic regimen/100ml of distilled water [7]. Parents were euthanized after newborns were weaned and the mice in the study groups were also euthanized at old age or upon onset of clinical symptoms such as severe weight loss, tremors, lethargy, splenomegaly and/or onset of hind limb paralysis by placing them in an approved chamber and applying carbon dioxide followed by a secondary method of cervical dislocation as per AVMA guidelines. Harvested tissues from study groups include blood, thymus, spleen, lymph nodes, cecum, colon, and feces.

## Viral load

The 15F Assay was used to determine infection and the viral load of murine blood samples and were provided by Dr. P. K.Y. Wong of the University of Texas M.D. Anderson Cancer Center, Smithville, Texas. 15F cells are nontransformed, nonproducer sarcoma-positive, leukemia-negative cell line derived from mouse thymus bone marrow cell line (TB) infected with Moloney murine sarcoma virus. Cultures infected with temperature sensitive (ts) mutants were incubated at permissive temperature of 34˚C to produce foci from both division of initially infected cells and recruitment of neighboring cells in subsequent rounds of replication [10,11]. Cells were incubated at 34˚C overnight in 60 mm culture plates containing 4ml of Dulbecco Modified Eagle Medium (DMEM) containing polybrene, 3% heat-inactivated newborn calf serum, and 1% penicillin/streptomycin. Serial dilutions of 200µl blood samples were added to 800µl polybrene media for a $10^1$ dilution. 200µl from this solution was pipetted into another 800µl of polybrene media for a $10^2$ dilution. This was repeated for $103–10^7$ dilutions. 500µl in duplicates of each of these concentrations were used in inoculating the 15F cells that were grown overnight for 40 minutes. Polybrene media were discarded and 4ml of fortified DMEM with 6% fetal calf serum, 4% newborn calf serum and 1% penicillin/streptomycin were added to each plate. Cells grew to 90–100% confluence by day 5 or 6 with media change on day 3, foci were counted and viral load calculated [8]. Hematoxylin and Eosin stained cross-sections of harvested tissues were also prepared.

## 16S rRNA gene sequencing

DNA was extracted and quantified using MoBio Powerfecal DNA isolation kit following manufacturer's instructions. This was followed by PCR amplification using Illumina iTag Polymerase Chain Reactions. PCR products were pooled, purified, denatured and loaded on Illumina

MiSeq V2 500 cycle kit cassette with 16S rRNA library sequencing primers and set for 250 base paired end reads. Raw paired-end reads were merged to create consensus sequences and then quality filtered using USEARCH [12]. Chimeric sequences were identified and filtered using Quantitative Insights Into Microbial Ecology (QIIME) software package (version 1.9.1) [13] combined with the USEARCH algorithm. A total of 403877 sequences were obtained after quality filtering and chimera checking. High quality sequence data was obtained for all group samples processed (n = 5) and sequencing depth of quality-filtered data ranged from 5840 to 52094 sequences per sample. Operational taxonomic units were subsequently picked using QIIME combined with the USEARCH algorithm, and taxonomy assignment was performed using Greengenes [14] as the reference database. Alpha and beta diversity analyses were performed using QIIME. Chao1 was used as the algorithm to calculate the alpha diversity. The analysis of similarities (ANOSIM) method was used to calculate the statistical significance of the beta diversity. Taxonomic features with differential abundance were further summarized using linear discriminant analysis effect size [15] for group comparisons.

## Results

### 1) Cecal microbiome diversity and taxa differential abundance analyses

Antibiotic treatment significantly decreased alpha diversity and altered overall gut microbiome diversity irrespective of ts1 infection (Fig 1A). Unweighted and weighted beta diversity analysis further confirms a significant population shift (ANOSIM p-value = 0.001) (Fig 1B and 1C). Relative abundance of bacterial phyla when quantified for the different groups show a notable change compared to control. Phylum Bacteroidetes relative abundance increased four-fold (4X) in ABXts1, ABX and increased 2X in ts1 compared to control. Phylum Proteobacteria relative abundance increased 65-fold in ABXts1, 75X in ABX and 2.8X in ts1 compared to control. Phylum Verrucomicrobia relative abundance increased 54X in ABXts1, 13X in ABX and there was no change in ts1 compared to control. On the other hand, Phylum Firmicutes relative abundance decreased 95% in ABXts1, 97% in ABX and 20% in ts1 group compared to control. Presence of Phylum Deferribacteres was negligible in both ABXts1 and ABX groups and decreased by 15% in ts1 group compared to control (Fig 1D). In these examples, both ABXts1 and ABX groups presented similar effects on the relative abundances of the selected bacterial phyla. Linear discriminant analysis further identified significantly enriched bacteria at taxonomic levels in different experimental groups (Fig 1E and 1F). For example, Gammaproteobacteria (class), Enterobacteriales (order), *Enterobacteriaceae* (family), *Porphyromonadaceae* (family), and *Parabacteroides* (genus) were more abundant in the ABXts1 group. Erysipelotrichi (class), Erysipelotrichales (order), *Erysipelotrichaceae* (family), *Bacteroidaceae* (family), and *Bacteroides* (genus) were more abundant in the ABX group. *Prevotellaceae* (family) and *Prevotella* (genus) were more enriched in the ts1 group. Clostridia (class), Mollicutes (class), *Lachnospiraceae* (family), *Peptococcaceae* (family), *Ruminococeae* (family), *Roseburia* (genus), *Mucispirillum* (genus), *Anaeroplasma* (genus), and *Coprococcus* (genus) were more enriched in the Ctrl group (Fig 1F).

### 2) Viral pathogenesis and ABX treatment

In the established ts1 model for perinatal infection, viral transmission and accompanying clinical symptoms were highly reproducible [8,9]. Treatment with ABX did not decrease viral transmission or symptom expression between ABXts1 and ts1 groups. ABXts1 (n = 6) group average plasma ts1 viral titer was 1.69 x 10$^5$ FFU/ml and ts1 (n = 6) group was 1.24 x 10$^5$ FFU/ml. Ctrl (n = 6) and ABX (n = 6) groups were negative for ts1 viral particles. Antibiotic treatments did not reduce ts1 viral load for the ABXts1 group as the titers were similar to the ts1

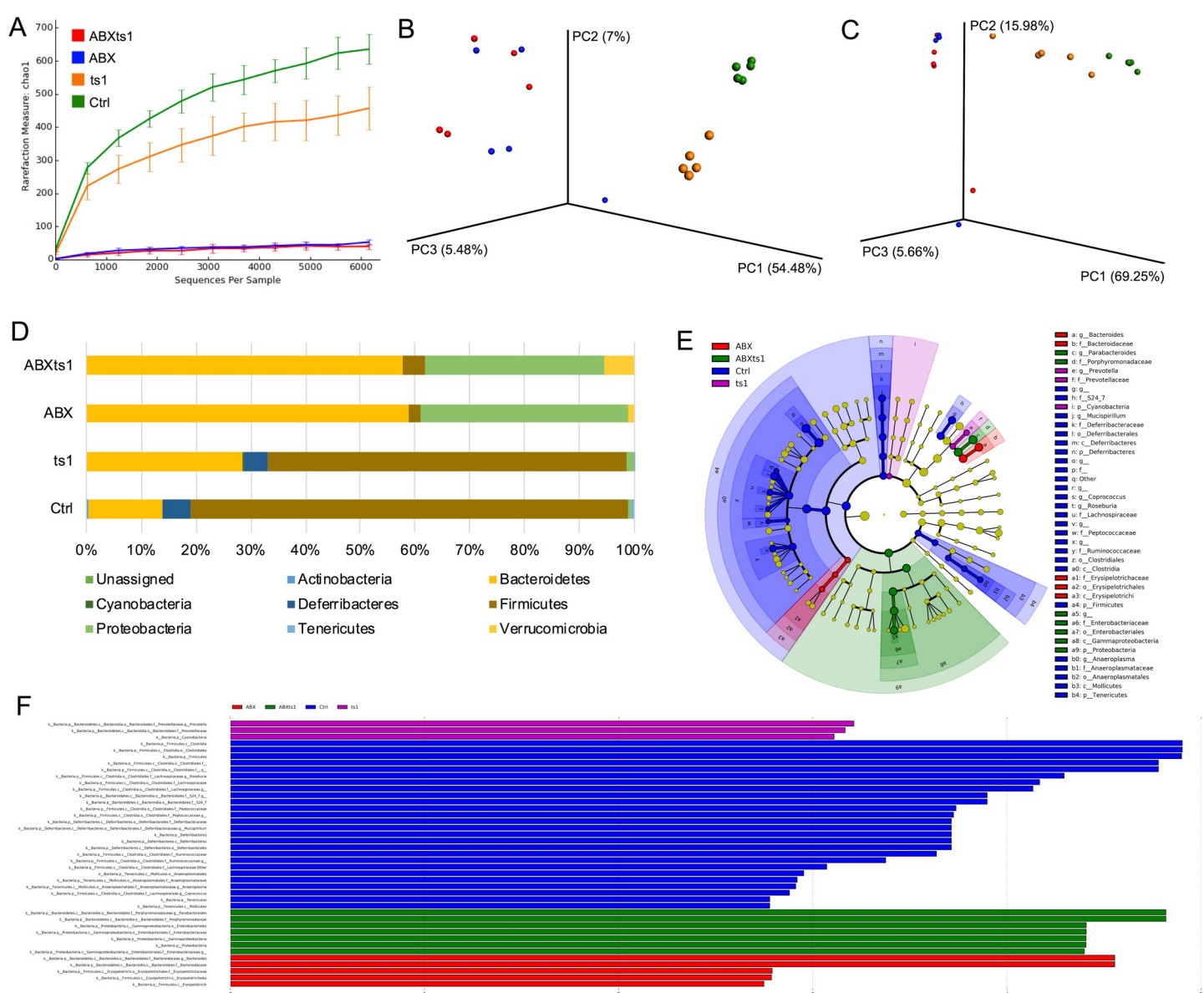

**Fig 1. Summary of cecal microbiome diversity and taxa differential abundance analyses.** (Fig 1A) Average α-diversity. All values are expressed as mean ± standard deviation. (Fig 1B) Unweighted β-diversity (ANOSIM p-value = 0.001). (Fig 1C) Weighted β-diversity (ANOSIM p-value = 0.001); (Fig 1D) Relative abundance of bacterial phyla. (Fig 1E) Cladogram showing enriched bacterial taxa with their phylogenetic relationships. (Fig 1F) LEfSe plot showing enriched bacterial taxa in each group. Bacterial taxa that have a linear discriminant analysis (LDA) score greater than 2.0 are considered to be significantly differential.

group which was not treated with antibiotics (Fig 2A). There were significant weight losses among the groups compared to Ctrl. TTEST based analysis, ABXts1 (n = 25) lost 28.2% (p < (0.05) = 6.9x10$^{-15}$); ts1 (n = 25) lost 16.5% (p < (0.05) = 1.42x10$^{-4}$) and ABX (n = 50) lost 11.2% (p < (0.05) = 6.03x10$^{-5}$) of their average body weights compared to Ctrl (n = 79), ABXts1 lost the most among all the groups. In general, infected groups lost the most weight (Fig 2B). Weight increases in spleen, thymus and mesenteric lymph node (MLN) confirmed classic manifestations of ts1 infection and were observed in ABXts1 and ts1 groups. ABXts1 (0.767g) and ts1 (0.800g) average spleen weight was over six-fold (6X) increase compared to Ctrl (0.0116gm) (p < (1.56x10$^{-6}$ and 7.56x10$^{-3}$ respectively). Spleen weight average for ABX treated group (0.091g) was

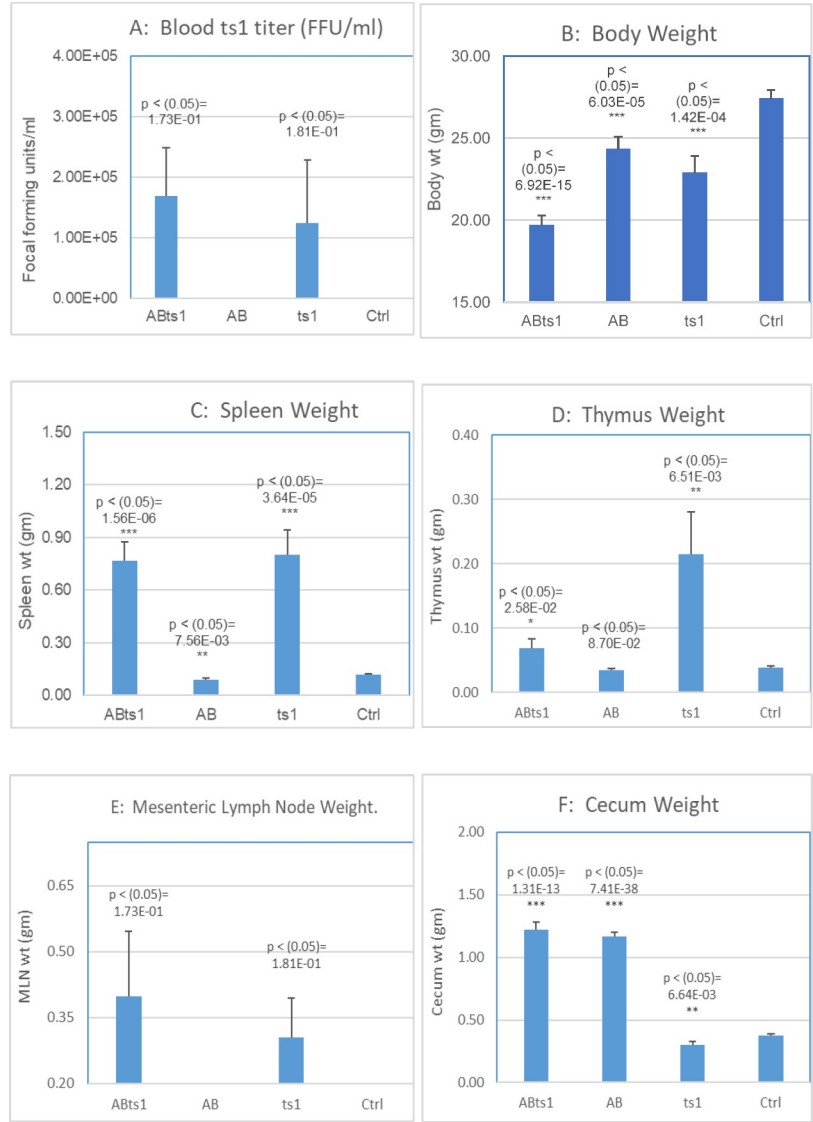

**Fig 2. Antibiotic independent viral growth.** Average peripheral blood ts1 titer for all groups (Fig 2A) were measured and compared to uninfected control. Average body weights (Fig 2B), spleen weights (Fig 2C) thymus weights (Fig 2D), mesenteric lymph node weights (Fig 2E) and cecum weights (Fig 2F) in ABXts1, ABX, and ts1 groups were measured and compared to control (Ctrl).

similar to Ctrl group (0.116g) (1X) (Fig 2C). Thymus average weight for ABXts1 group (0.069gm) increased almost two-fold (1.8X) and ts1 (0.215g) increased over five-fold (5.5X) compared to Ctrl group (0.039g) (Fig 2D). Average weight of MLN for ABXts1 group (0.399g) increased over 390-fold and ts1 group (0.305g) increased 300-fold compared to Ctrl group (0.000mg) (Fig 2E). Average weight of cecum for ABXts1 (1.220g) and ABX (1.164g) represents an average of 3.5X increase compared to Ctrl (0.375g) and ts1 (0.303g) (Fig 2F).

## 3) ts1-mediated pathology is not altered by ABX

The anatomy and histology of spleen and thymus from ABX treated group (Fig 3C and 3G) were comparable to the Ctrl group (Fig 3A and 3E). The matrix of the spleen contains discrete

distribution of red and white pulp (Fig 3E and 3G). However, ts1 and ABXts1 groups spleen and thymus were morbidly enlarged (Fig 3B and 3D) and the spleen matrix of ts1 and ABXts1 showed architectural disarray with several merged white pulp regions and inconsistent red pulp regions generally observed in lymphoma presentation (Fig 3F and 3H). Thymus for Ctrl and ABX groups retained their lobulated architecture with prominent basophilic outer cortex appearance (purple) (Fig 3J and 3L). Histology sections for ts1 and ABXts1 groups depicted loss of lobular structure and massive fatty infiltration into both the cortex and medulla (lighter purple) (Fig 3K and 3M). Another notable variation was the gross appearance of the cecum with Ctrl and ts1 groups similar in appearance and weight (Fig 3A and 3B) compared to ABXts1 and ABX (Fig 3C and 3D) that were over 3X their weight and volume. For the colon, cross-section of the inner circular and outer longitudinal muscularis were expanded for ABX and ABXts1 groups (Fig 3Q and 3R) compared to Ctrl and ts1 groups (Fig 3N and 3P).

## Discussion

Normal commensal microbes play a crucial role in the maintenance of health and protection from disease. Oral antibiotic use has been shown to cause dysbiosis of GI microbiota by promoting disruption of resident bacteria leading to the emergence and expansion of non-resident pathogens with a rapid decrease in taxonomic richness, diversity and evenness of GI microbes [16–20]. Alterations in normal commensal microbes decrease viral transmission and subsequent pathogenesis in some but not all enteric viral infections [6,7]. The current study evaluated the role of the intact and altered murine microbiome on ts1 transmission and the development of neoplastic changes as well as the effects of the virus itself on the microbiome. Results from this study show that bacterial phyla alteration were observed in all test groups to varying degrees compared to control.

Antibiotic treatment had profound effects on microbiota with extensive loss of alpha diversity and altered beta diversity in the gut microbiome. However, these changes did not alter ts1 retroviral transmission, viral load, spleen size or impact neoplastic changes. Taken together, these results suggest that ts1 can alter the gut microbiome to some extent but viral transmission and subsequent neoplasia in splenic and thymic tissue was not altered by bacterial dysbiosis. Also, the presentations normally associated with long term antibiotic treatment were not hindered by ts1 infection such as morbidly enlarged cecum (Fig 3C and 3D). This contrasts with the effects of the gut microbiome on transmission of other viral pathogens, such as Murine Mammary Tumor Virus (MMTV), where infection cannot be established with gut dysbiosis [7].

When relative abundances were quantified for the different groups, the degree of change was notable. Distinct significant changes occurred in selected phyla in their relative abundances based primarily on the antibiotic treatment rather than the viral infection. Groups receiving antibody treatment such as ABXts1 and ABX, phyla Bacteroidetes, Proteobacteria and Verrucomicrobia increased significantly compared to control while moderate increases to no change was observed in the ts1 infected group. Likewise, antibiotic treated groups ABXts1 and ABX show significant decreases in bacterial relative abundance for phyla Firmicutes and Deferribacteres of over 95% compared to control, while decreases for the ts1 infected group were only 15 to 20%.

Regarding the bacterial changes, unique enriched taxa for each group using cladogram (Fig 1E) and LEfSe (Fig 1F) plots showed that control group (Ctrl) was enriched in Clostridia (class), Mollicutes (class), *Lachnospiraceae* (family), *Peptococcaceae* (family), *Ruminococeae* (family), *Roseburia* (genus), *Mucispirillum* (genus), *Anaeroplasma* (genus) and *Coprococcus* (genus). These taxa are notable for maintaining microbiome stability, protection and as

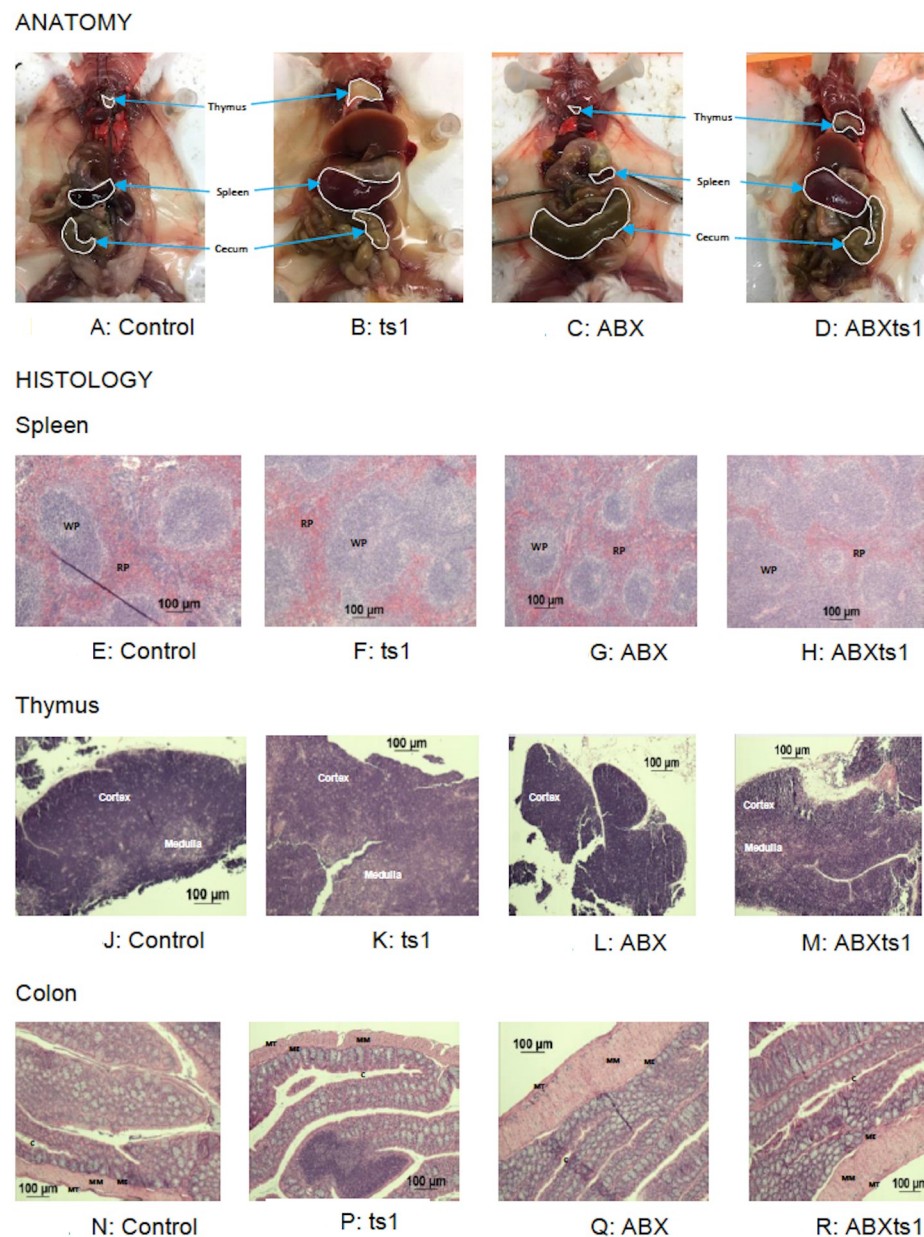

**Fig 3. Anatomic and histologic changes due to viral infection and antibiotic treatment.** Anatomic changes showing spleen, thymus and cecum from Control (Fig 3A), ts1 infected (Fig 3B), ABX treated (Fig 3C) and ABXts1 treated and infected (Fig 3D) animals. Spleen architecture with normal distribution of discrete red pulp (RP) and white pulp (WP) regions for control (Fig 3E) and ABX treated (Fig 3G). Mice infected with ts1 (Fig 3F) and ABXts1 treated and infected animals (Fig 3H), show disorganized red and white pulp regions. Thymic capsule, cortex and medulla among the various groups: control (Fig 3J) and ABX treated (Fig 3L) show intact capsule and distinct areas of the cortex (deep purple) and medulla (lighter purple) of thymus; ts1 infected (Fig 3K) and ABXts1 treated and infected animals (Fig 3M) were fragile, with ill-defined capsule and diffused cortex and medulla regions. Colon (Fig 3N, 3P, 3Q and 3R): Normal muscularis tunics (MT), muscularis externa (ME), muscularis mucosae (MM), crypts (C) of colon in control (Fig 3N) and ts1 infected (Fig 3P) animals. Expanded or broad layers of muscularis tunics (MT), muscularis externa (ME) and muscularis mucosae (MM) with narrowed crypts (C) of colon in ABX treated (Fig 3Q) and ABXts1 treated and infected (Fig 3R) animals.

sources of energy. Unique enriched taxa for ts1 group include *Prevotellaceae* (family) and *Prevotella* (genus); ABXts1 group unique enriched taxa include Gammaproteobacteria (class), Enterobacteriales (order), *Enterobacteriaceae* (family), *Porphyromonadaceae* (family) and *Parabacteroides* (genus); ABX group enriched taxa include Erysipelotrichi (class), Erysipelotrichales (order), *Erysipelotrichaceae* (family), *Bacteroidaceae* (family) and *Bacteroides* (genus) (Fig 1F). Some studies have shown that composition of long-term diets can lead to clustering of distinct microbes and other studies have shown that expanding sample sizes or duration of experiment will show a gradient continuum rather than distinct clusters for the varied diet [21–24]. Our experiment has accounted for any diet induced variability through provision of a standard long-term diet for all groups with an added step of autoclaving the diets for antibiotic treated groups, preventing the introduction of external microbes.

In conclusion, the murine model of intraperitoneal and postpartum ts1 infection does not depend on the presence of either normal or altered gastrointestinal microbes. The development of neoplasia also is not affected by alterations in the gut microbiome. Microbiome unique taxa enrichment for the treatment and infection groups, and the relative abundance of bacterial phyla for each group may present as acceptable biomarkers for ts1 infection and antibiotic driven microbiome diversity.

## Supporting information

**S1 Fig. Microbiome antibiotic–ts1 flowchart.** Experimental design illustrating the inoculation, transmission of ts1 virus and/or initiation of antibiotic treatment based on the grouping. (TIF)

**S1 Table. Microbiome diversity: Prominent genera comparisons showing average relative abundance of bacteria taxa identified within each respective experimental group.** Clostridiales comprise a substantial portion of the microbiome within groups not treated with antibiotics, ts1 and control. (TIF)

**S2 Table. Tissue weights: Pathological changes associated with ABXts1, ABX and ts1 infection and/or treatment compared to Ctrl.** Independent analogies can be inferred from the effects of the virus or the effects of the antibiotic treatment. (TIF)

## Acknowledgments

This work was supported by F. M. Douglas Foundation of St. Vincent Medical Center.

## Author Contributions

**Conceptualization:** Henry Okonta, Joan Duggan.

**Formal analysis:** Henry Okonta, Xi Cheng, Ritu Chakravarti, Joan Duggan.

**Investigation:** Henry Okonta.

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
