## [Decision Letter · Decision Letter 0]

13 Jul 2021

PONE-D-21-14921

Effects of antibiotic treatment on microbiota, viral transmission and viral pathogenesis of MoMuLV ts1 infected BALB/c mice

PLOS ONE

Dear Dr. Duggan,

Thank you for submitting your manuscript to PLOS ONE. After careful consideration, we feel that it has merit but does not fully meet PLOS ONE’s publication criteria as it currently stands. Therefore, we invite you to submit a revised version of the manuscript that addresses the points raised during the review process.

We look forward to receiving your revised manuscript.

Kind regards,

Mehreen Arshad, M.D.

Academic Editor

PLOS ONE

Journal Requirements:

2. As part of your revisions please update your Methods to address the following: (1) if applicable: administration of anesthetics, analgesics, antibiotics, etc. during the study; (2) if applicable: unanticipated adverse events involving animals; and (3) monitoring parameters. Thank you for your cooperation in this matter

Reviewers' comments:

Reviewer's Responses to Questions

**Comments to the Author**

1. Is the manuscript technically sound, and do the data support the conclusions?

Reviewer #1: No

Reviewer #2: Yes

Reviewer #3: Yes

2. Has the statistical analysis been performed appropriately and rigorously? 

Reviewer #1: No

Reviewer #2: Yes

Reviewer #3: I Don't Know

3. Have the authors made all data underlying the findings in their manuscript fully available?

Reviewer #1: No

Reviewer #2: Yes

Reviewer #3: Yes

4. Is the manuscript presented in an intelligible fashion and written in standard English?

Reviewer #1: No

Reviewer #2: Yes

Reviewer #3: Yes

5. Review Comments to the Author

Reviewer #2: 1.The methods section does not detail the isolation and preparation of the bacterial population for the 16S RNA-seq. Can the authors provide such details in the methods section?

2.Can the authors add statistical analysis to the figure 2. It is not clear how many replicates are used in the experiment and the statistical method for the group comparisons

3.Is there any correlation between ts1 infection or antibiotic exposure and the observed neoplasms in the study. Also can the microbial dysbiosis be correlated with the observed neoplasms in the study.

There are a number of grammatical errors which should be considered by the authors:

Below are a few examples:

Line 28: Transmission and pathophysiology of ts1 infection were (not was) not significantly…

Line 62: Sentence looks incomplete: analyses at terminal end stage for infected and age comparable uninfected mice.in

Line 62: terminal end-stage

Line 74: labeled not labelled

Line 199: Cross section should be cross-section

Line 227: alterations in normal commensal microbes decrease( not decreases) viral transmission

Reviewer #3: This study indicated the finding that transmission and pathophysiology of ts1 infection was not significantly altered by the microbial composition of the GI tract. This is an interesting study that can provide a reference for studies on Moloney Murine Leukemia Virus.

Major comments:

-Why choose temperature sensitive mutant 1 to infect animals?

-There is a lot of information describing the infection program, and it is difficult to quickly understand the author's infection program clearly. Tables or diagrams may be displayed more clearly.

-What are the changes in the cecal microbiome diversity before and after the virus infection?

-In figure 2C-F, is a significant difference analysis performed? If so, please mark the p-value.

-fig 2A, please comment on the big difference in Peripheral blood ts1 titer in the ABXts1 group. The ts1 group is the same, please comment.

-Except for the viral load in Peripheral blood and body weight, what other data can assess the pathogenicity of the virus?

Minor comments:

- define all the acronyms, like “15F” in line 92

-What is mean of “in” in line 62?

6. PLOS authors have the option to publish the peer review history of their article (what does this mean?). If published, this will include your full peer review and any attached files.

Reviewer #1: No

Reviewer #2: No

Reviewer #3: No

---

## [Author Response · Author response to Decision Letter 0]

7 Sep 2021

Dear Reviewers, Thank you for allowing us to submit a revised version of the manuscript Effects of antibiotic treatment on microbiota, viral transmission and viral pathogenesis of MoMuLV ts1 infected BALB/c mice PONE-D-21-14921 and address the points raised during the review process. We look forward to hearing from you soon. Please do not hesitate to contact me for any additional questions and concerns. 

Sincerely, 

Joan M. Duggan, MD, FACP, FIDSA, AAHIVMS

University of Toledo Health Science Campus

Senior Associate Dean for Faculty Affairs and Development

Principle Investigator/ Medical Director Ryan White Parts C/D

Phone: 419 – 383 – 4200

Fax (Academic Office): 419 – 383 – 2950

joan.duggan@utoledo.edu

---

## [Decision Letter · Decision Letter 1]

9 Dec 2021

Effects of antibiotic treatment on microbiota, viral transmission and viral pathogenesis of MoMuLV ts1 infected BALB/c mice

PONE-D-21-14921R1

Dear Dr. Duggan,

We’re pleased to inform you that your manuscript has been judged scientifically suitable for publication and will be formally accepted for publication once it meets all outstanding technical requirements.

Kind regards,

Mehreen Arshad, M.D.

Academic Editor

PLOS ONE

Additional Editor Comments (optional):

Reviewers' comments:

Reviewer's Responses to Questions

**Comments to the Author**

1. If the authors have adequately addressed your comments raised in a previous round of review and you feel that this manuscript is now acceptable for publication, you may indicate that here to bypass the “Comments to the Author” section, enter your conflict of interest statement in the “Confidential to Editor” section, and submit your "Accept" recommendation.

Reviewer #1: All comments have been addressed

Reviewer #2: All comments have been addressed

2. Is the manuscript technically sound, and do the data support the conclusions?

Reviewer #1: Yes

Reviewer #2: Yes

3. Has the statistical analysis been performed appropriately and rigorously? 

Reviewer #1: Yes

Reviewer #2: Yes

4. Have the authors made all data underlying the findings in their manuscript fully available?

Reviewer #1: Yes

Reviewer #2: Yes

5. Is the manuscript presented in an intelligible fashion and written in standard English?

Reviewer #1: Yes

Reviewer #2: Yes

6. Review Comments to the Author

Reviewer #1: Authors addressed my comments therefore my recommendation is positive for this manuscript. Authors advised further to check the proofreading of the manuscript carefully. Specially the affiliation and other scientific content.

Reviewer #2: The authors have provided adequate response to the review questions and have addressed the grammatical errors present in the initial manuscript.

7. PLOS authors have the option to publish the peer review history of their article (what does this mean?). If published, this will include your full peer review and any attached files.

Reviewer #1: **Yes: **Pankaj Bhatt

Reviewer #2: No

---

## [Editor Report · Acceptance letter]

20 Dec 2021

PONE-D-21-14921R1 

Effects of antibiotic treatment on microbiota, viral transmission and viral pathogenesis of MoMuLV ts1 infected BALB/c mice 

Dear Dr. Duggan:

I'm pleased to inform you that your manuscript has been deemed suitable for publication in PLOS ONE. Congratulations! Your manuscript is now with our production department. 

Kind regards, 

on behalf of

Dr. Mehreen Arshad 

Academic Editor

PLOS ONE